# A new scoring system with simple preoperative parameters as predictors of early recurrence of pancreatic ductal adenocarcinoma

**Tomonari Shimagaki**[1]*, **Keishi Sugimachi**[1], **Yohei Mano**[1], **Takahiro Tomino**[1],
**Emi Onishi**[1], **Yuichiro Nakashima**[2], **Masahiko Sugiyama**[2], **Manabu Yamamoto**[2],
**Masaru Morita**[2], **Mototsugu Shimokawa**[3], **Tomoharu Yoshizumi**[4], **Yasushi Toh**[2]

1 Department of Hepatobiliary and Pancreatic Surgery, National Hospital Organization Kyushu Cancer Center, Fukuoka, Japan, 2 Department of Gastroenterological Surgery, National Hospital Organization Kyushu Cancer Center, Fukuoka, Japan, 3 Department of Biostatistics, Yamaguchi University Graduate School of Medicine, Yamaguchi, Japan, 4 Department of Surgery and Science, Graduate School of Medical Sciences, Kyushu University, Fukuoka, Japan

* rgwsn6911@gmail.com

**Data Availability Statement:** The data of this paper have been uploaded to the public repository of the

## Abstract

### Background

Pancreatic ductal adenocarcinoma (PDAC) often recurs early after radical resection, and such early recurrence (ER) is associated with a poor prognosis. Predicting ER is useful for determining the optimal treatment.

### Methods

One hundred fifty-three patients who underwent pancreatectomy for PDAC were divided into an ER group (n = 54) and non-ER group (n = 99). Clinicopathological factors were compared between the groups, and the predictors of ER and prognosis after PDAC resection were examined.

### Results

The ER group had a higher platelet count, higher platelet-to-lymphocyte ratio (PLR), higher preoperative CA19-9 concentration, higher SPan-1 concentration, larger tumor diameter, and more lymph node metastasis. The receiver operating characteristic (ROC) curve analysis identified cut-off values for PLR, carbohydrate antigen 19–9 (CA19-9), SPan-1, and tumor diameter. In the multivariate analysis, a high PLR, high CA19-9, and tumor diameter of >3.1 cm were independent predictors of ER after resection (all p < 0.05). When the parameter exceeded the cut-off level, 1 point was given, and the total score of the three factors was defined as the ER prediction score. Next, our new ER prediction model using PLR, CA19-9 and tumor diameter (Logit(p) = 1.6 + 1.2 × high PLR + 0.7 × high CA19-9 + 0.5 × tumor diameter > 3.1cm) distinguished ER with an area under the curve of 0.763, a sensitivity of 85.2%, and a specificity of 55.6%.

Open Science Framework (OSF). The URL below is the URL for that OSF. https://osf.io/gyx4h/.

**Funding:** The author(s) received no specific funding for this work.

**Competing interests:** The authors have declared that no competing interests exist.

**Abbreviations:** PDAC, pancreatic ductal adenocarcinoma; ER, early recurrence; PLR, platelet-to-lymphocyte ratio; CA19-9, carbohydrate antigen 19–9; ROC, receiver operating characteristic; NLR, neutrophil-to-lymphocyte ratio; mGPS, modified Glasgow Prognostic Score; NAT, neoadjuvant therapy; R, resectable; BR-A, borderline resectable with artery involvement; BR-PV, borderline resectable with portal vein involvement; UR-LA, unresectable with locally advanced factors; UR-M, unresectable with metastatic factors; GEM, gemcitabine; nab-PTX, nanoparticle albumin bound paclitaxel; CONUT, controlling nutritional status; AUC, area under the curve; DFS, disease-free survival; OS, overall survival; SSLR, stratum specific likelihood ratio.

## Conclusions

ER after resection of PDAC can be predicted by calculation of a score using the preoperative serum CA19-9 concentration, PLR, and tumor diameter.

## Introduction

Pancreatic ductal adenocarcinoma (PDAC) is a disease with a poor prognosis, and early recurrence (ER) within a short period after surgical resection frequently occurs because of the aggressive nature of the disease. Advances in neoadjuvant chemotherapy and postoperative adjuvant chemotherapy have improved patient survival after surgical resection of pancreatic cancer [1–3]. Although radical resection is considered a curative treatment, the 5-year survival rate after curative resection is only 15% to 20% [4]. However, distant metastases are often detected early after resection [5], and some patients must discontinue adjuvant chemotherapy. Therefore, the ability to predict which patients will develop rapidly progressive disease using preoperative parameters would be valuable.

Previous studies have investigated factors related to the uncontrollable behavior of PDAC [6]. In particular, the serum carbohydrate antigen 19–9 (CA19-9) concentration is known to correlate with the postoperative prognosis, and its association with postoperative ER has also attracted attention [7]. The preoperative serum CA19-9 concentration is reportedly an independent factor that determines postoperative ER, and it has been perceived as an indicator of a poor postoperative prognosis [8]. As we previously reported, other studies have addressed the association of tumor immunonutrient factors, such as the neutrophil-to-lymphocyte ratio (NLR) and modified Glasgow Prognostic Score (mGPS), with postoperative survival [9, 10]. Few studies of patients who have undergone pancreatic cancer surgery have considered perioperative factors such as preoperative nutritional evaluation and radiological imaging. Therefore, predicting ER is important for determining the optimal treatment while taking perioperative factors into account.

The significance of neoadjuvant therapy (NAT) for such difficult-to-control PDAC has been discussed [11]. The purpose of NAT is to achieve more reliable R0 resection and control latent distant metastasis preoperatively, which is expected to have a further prognostic effect on resectable PDAC [12]. In addition, new combinations of anticancer drugs that are expected to affect pancreatic cancer have been developed [13, 14]. Thus, pancreatic cancer treatment has entered a phase of major change in which it is possible to select an effective and appropriate treatment based on the tumor profile of each individual patient. Selection of an effective treatment requires preoperative determination of the tumor characteristics, including the potential for ER within 1 year after the operation.

The aim of this retrospective study was to establish a prognostic scoring system based on preoperative biomarkers to identify ER within 1 year after radical resection of PDAC.

## Materials and methods

### Study design

This retrospective study was conducted using prospectively collected and maintained data from January 2014 to December 2020 at the Kyushu Cancer Center. This study included 153 consecutive patients with cytohistologically proven PDAC underwent pancreatectomy for curative intent. The cases those resulted in non-resection were excluded from the analysis. In

this study, we used the 7th edition of the classification of pancreatic carcinoma by Japan Pancreas Society. When classified as initial resectability, resectable (R) / borderline resectable with artery involvement (BR-A) / borderline resectable with portal vein involvement (BR-PV) / unresectable with locally advanced factors (UR-LA) / unresectable with metastatic factors (UR-M) were 131/7/6/4/5 cases (Table 1). This study included 5 Stage IV (UR-M) cases those underwent conversion surgery after systemic chemotherapy. These included 2 cases of

**Table 1. Comparative analysis of clinicopathological parameters between ER group and non-ER group.**

| Variable | Total (n = 153) | non-ER group (n = 99) | ER group (n = 54) | P value |
|---|---|---|---|---|
| Gender (male/female) | 90/63 | 55/44 | 35/19 | 0.2661 |
| Age (years)[#] | 68.8±0.8 | 69.3±1.0 | 67.9±1.4 | 0.4217 |
| BMI (kg/m$^2$)[#] | 22.2±0.2 | 22.1±0.3 | 22.2±0.4 | 0.9162 |
| DM (yes/no) | 72/81 | 46/53 | 26/28 | 0.8420 |
| Initial resectability (R/BR-A/BR-PV/UR-LA/UR-M) | 131/7/6/4/5 | 90/2/3/1/3 | 41/5/3/3/2 | 0.0813 |
| Platelet (×10$^4$/μL)[#] | 22.6±0.6 | 21.5±0.8 | 24.6±1.0 | **0.0159** |
| Serum albumin (g/dL)[#] | 3.9±0.1 | 3.9±0.1 | 3.9±0.1 | 0.8517 |
| Total bilirubin (mg/dL)[#] | 0.8±0.1 | 0.8±0.1 | 0.7±0.1 | 0.7233 |
| P-Amylase (U/L)[#] | 52.3±6.6 | 47.1±8.3 | 61.6±11.1 | 0.2971 |
| NLR[#] | 2.3±0.1 | 2.20±0.12 | 2.46±0.17 | 0.2056 |
| PLR[#] | 0.9±0.1 | 0.78±0.05 | 0.99±0.06 | **0.0107** |
| LMR[#] | 4.9±0.3 | 5.17±0.33 | 4.52±0.45 | 0.2417 |
| mGPS[#] | 0.26±0.04 | 0.27±0.05 | 0.24±0.07 | 0.7251 |
| TLC[#] | 1574.3±44.9 | 1563.3±75.8 | 1594.3±75.8 | 0.7425 |
| PNI[#] | 46.5±0.4 | 46.4±0.5 | 46.6±0.7 | 0.7388 |
| CONUT score[#] | 1.9±0.1 | 1.9±0.2 | 1.8±0.2 | 0.7418 |
| CAR[#] | 0.11±0.02 | 0.127±0.028 | 0.082±0.038 | 0.3503 |
| Preoperative chemotherapy (yes/no) | 55/98 | 32/67 | 23/31 | 0.2059 |
| Tumor location (Ph/Pb/Pt) | 77/46/30 | 45/36/18 | 32/10/12 | 0.2118 |
| CEA (ng/mL)[#] | 5.2±1.0 | 5.5±1.2 | 4.4±1.7 | 0.5798 |
| CA19-9 (U/mL)[#] | 274.4±45.8 | 164.8±54.2 | 430.7±73.3 | **0.0041** |
| DUPAN-2 (U/mL)[#] | 435.0±91.5 | 355.4±113.7 | 581.0±153.9 | 0.2400 |
| SPan-1 (U/mL)[#] | 47.0±7.1 | 32.9±8.6 | 72.6±11.6 | **0.0068** |
| Operative time (min)[#] | 329.6±9.5 | 313.4±11.7 | 359.4±15.8 | **0.0203** |
| Blood loss (g)[#] | 312.5±25.2 | 265.9±30.5 | 423.3±41.3 | **0.0026** |
| Peritoneal lavage cytology (CY) X/0/1 | 10/143/0 | 6/93/0 | 4/50/0 | 0.7474 |
| Maximum tumor size (cm)[#] | 2.8±0.1 | 2.7±0.1 | 3.2±0.2 | **0.0195** |
| Lymph node metastasis (yes/no) | 94/59 | 54/45 | 40/14 | **0.0177** |
| Diffrentiation (well, moderate/poorly) | 69/84 | 49/50 | 20/34 | 0.1389 |
| R0 resection (yes/no) | 148/5 | 99/ 0 | 49/ 5 | **0.0021** |
| pStage 1/2/3/4 | 18/93/26/16 | 13/59/17/10 | 5/34/9/6 | 0.9078 |
| Postoperative complication CD (0-1/≥2) | 83/70 | 51/48 | 32/22 | 0.3582 |
| Adjuvant chemotherapy (yes/no) | 123/30 | 80/19 | 43/11 | 0.8607 |

ER, early recurrence; BMI, body mass index; DM, diabetes mellitus; R, resectable; BR-A, borderline resectable with artery involvement; BR-PV, borderline resectable with portal vein involvement; UR-LA, unresectable with locally advanced factors; UR-M, unresectable with metastatic factors; NLR, neutrophil-to-lymphocyte ratio; PLR, platelet-to-lymphocyte ratio; LMR, lymphocyte-to-monocyte ratio; mGPS, modified Glasgow Prognostic Score; TLC, total lymphocyte count; PNI, prognostic nutritional index; CONUT, Controlling Nutritional Status; CAR, C-reactive protein/albumin ratio; CEA, carcinoembryonic antigen; CA19-9, carbohydrate antigen 19–9; CD, Clavien–Dindo classification.

[#]Data are expressed as mean ± standard error.

Boldface p values are statistically significant.

peritoneal dissemination, 2 of liver metastasis, and 1 of paraaortic lymph node metastases. In cases of peritoneal dissemination, pancreatectomy was done after exploring laparoscopy to confirm dissemination disappeared. In other 3 cases, metasectomy was done with pancreatectomy.

Among these 153 patients, 54 patients who developed recurrence within 1 year after macroscopically radical surgery were included in the ER group, whereas 99 patients in whom recurrence did not occur within 1 year constituted the non-ER group. We compared the clinicopathological parameters between the two groups, and the predictors of ER and prognosis after resection of PDAC were examined. Postoperative complications were evaluated based on the Clavien–Dindo classification [15].

This study protocol complied with the ethical guidelines of human clinical research established by the Japanese Ministry of Health, Labour and Welfare as well as with the 1964 Helsinki declaration and its later amendments. The study was approved by the Ethics and Indications Committee of the National Hospital Organization Kyushu Cancer Center (No. 2019–54). Informed consent was obtained from all individual participants included in the study.

## Nutritional evaluation

Preoperative blood samples were obtained from all patients within 1 week before surgery. Blood tests in the preoperative treatment group also showed results just before surgery, and the NLR, platelet-to-lymphocyte ratio (PLR), and lymphocyte-to-monocyte ratio were calculated based on the preoperative blood values. The mGPS was calculated as previously described [16]. Briefly, for the mGPS, patients with both an elevated C-reactive protein concentration ($>0.5$ mg/dL) and low albumin concentration ($<3.5$ g/L) were assigned a score of 2, patients with only one of these biochemical abnormalities were assigned a score of 1, and patients with neither of these abnormalities were assigned a score of 0. The total lymphocyte count was also evaluated. The prognostic nutritional index was calculated as follows: $10 \times$ serum albumin concentration (g/dL) + $0.005 \times$ lymphocyte count (cells/mm$^3$). The C-reactive protein/albumin ratio was calculated by dividing C-reactive protein by serum albumin. The Controlling Nutritional Status (CONUT) score was calculated using the serum albumin concentration, peripheral lymphocyte count, and total cholesterol concentration: (a) albumin concentrations of 3.5, 3.0 to 3.49, 2.5 to 2.99, and $<2.5$ g/dL were scored as 0, 2, 4, and 6 points, respectively; (b) total lymphocyte counts of 1600, 1200 to 1599, 800 to 1199, and $<800$ cells/mm$^3$ were scored as 0, 1, 2, and 3 points, respectively; and (c) total cholesterol concentrations of 180, 140 to 179, 100 to 139, and $<100$ mg/dL were scored as 0, 1, 2, and 3 points, respectively. The CONUT score was defined as the sum of (a), (b), and (c).

## Pathological evaluation

Pathological evaluation of surgical specimens was based on the tumor–node–metastasis (TNM) classification system of malignant tumors by the 7th edition of the classification of pancreatic carcinoma by Japan Pancreas Society. Evaluation of the excised margin was performed as follows. R0 resection was defined as the absence of tumor cells on the pancreatic resection margin, nerve plexus dissection margin, portal vein dissection surface, posterior dissection surface, and bile duct dissection margin. In contrast, R1 resection was defined as the presence of tumor cells on these margins. In all cases, rapid intraoperative pathological diagnosis of the pancreatic resection margin was conducted, and additional resection was performed when tumor cells were still present on the stump surface.

### Preoperative chemotherapy

According to the institutional protocols, 55 patients received modern combinatorial chemotherapy [including NAT with either a 5-fluorouracil-based chemotherapy (FOLFIRINOX/FOLFOX) or gemcitabine-based (GEM/nab-PTX) regimen or gemcitabine and S-1 combination therapy] with or without dose modifications as deemed appropriate by the treating oncologists. Chemotherapy was administered on a 2-week or 4-week cycle depending on the specific regimen. Total cycles of chemotherapy were counted based on the total number of cycles administered during induction chemotherapy. Two selected patients received chemoradiotherapy as NAT, and this was determined according to surgical/oncological recommendations in general based on the margin risk. In accordance with the institutional protocol, chemoradiotherapy consisted of photon/proton external beam therapy with a 50.4-Gy dose delivered in 28 daily fractions over 5 weeks using three-dimensional conformal or intensity-modulated techniques with concurrent radiosensitizing chemotherapy.

### Postoperative adjuvant chemotherapy

Postoperative adjuvant chemotherapy was introduced within 12 weeks after surgery and continued for 6 months. The chemotherapy regimen included GEM monotherapy, S-1 monotherapy, GEM and S-1 combination therapy, or GEM and nab-PTX combination therapy. No patients received postoperative radiation. After the operation, a blood examination including measurement of tumor markers (carcinoembryonic antigen, CA19-9, SPan-1, and DUPAN-2) and a computed tomography examination were done every 3 months for at least 5 years to screen for postoperative recurrence.

### Statistical analysis

The clinicopathological records of all 153 patients were collected and retrospectively reviewed. Differences between the ER group and non-ER group were assessed by the Mann–Whitney U test. Associations between variables were determined by Fisher's exact test or the $\chi^2$ test. The best cut-off value of each parameter was determined using receiver operating characteristic (ROC) curves and comparing the areas under the curves (AUCs). A stepwise multivariate analysis was conducted to identify parameters that significantly contributed to ER after pancreatectomy for PDAC.

Disease-free survival (DFS) and overall survival (OS) curves were calculated using the Kaplan–Meier method, and differences between groups were assessed using the log-rank test. DFS and OS in both the preoperative treatment group and the non-preoperative group showed years after resection. Univariate and multivariate logistic regression analyses were performed to identify independent determinants of ER. The stratum specific likelihood ratio (SSLR) is the probability of a given test result when the disease is present, divided by the probability of the same test result when the disease is absent. We determined these ratios by means of the formula $SSLR = (x1/n1)/(x0/n0)$, where x1 is the number of patients in the stratum with ER; n1 is the total number of patients with ER; x0 is the number of patients in the stratum without ER; n0 is the total number of patients without ER [17]. Statistical analyses were performed using GraphPad Prism, version 7.0 (GraphPad Software, San Diego, CA, USA) and JMP Pro 15.1 (SAS Institute, Cary, NC, USA). A p-value of $<0.05$ was considered statistically significant.

## Results

### Patient characteristics

The patients' characteristics are shown in Table 1. Their mean age was 68.8 years (range, 39–87 years). A preoperative nutritional assessment was performed using various indexes such as

the NLR, PLR and mGPS. The mean preoperative serum carcinoembryonic antigen and CA19-9 concentrations were 5.2 ± 1.0 ng/mL and 274.4 ± 45.8 mAU/mL, respectively. Preoperative chemotherapy was used in 55 of 153 patients, and postoperative adjuvant chemotherapy was used in 123 of 153 patients. In this study, when classified as initial resectability, R/BR-A/BR-PV/UR-LA/UR-M were 131/7/6/4/5 cases. 4 UR-LA cases and 5 UR-M cases were all conversion cases (Table 1). In this study, peritoneal lavage cytology (CY) X/0/1 was 10/143/0 cases (Table 1). This study did not include the cases with positive peritoneal cytology underwent only abdominal exploration.

Clavien–Dindo grade >2 postoperative complications occurred in 70 (45.6%) of the 153 patients, including pancreatic fistula (n = 43), delayed gastric emptying (n = 14), bile leakage (n = 1), ascites (n = 11), deep vein thrombosis (n = 11), delirium (n = 9), and wound infection (n = 23).

When OS was confirmed by stratifying by the presence or absence of preoperative treatment and postoperative adjuvant chemotherapy, the prognosis tended to be good in both groups that received chemotherapy although there were not statistically significant difference (S1A, S1B Fig in S1 File).

## Comparison between ER and non-ER groups

Comparison of the ER group (n = 54) and non-ER group (n = 99) showed that the platelet count, PLR, serum CA19-9 concentration, and SPan-1 concentration were significantly higher in the ER group than in the non-ER group (Table 1). Additionally, the tumor size was significantly larger and lymph node metastasis was significantly more severe in the ER group than in the non-ER group (Table 1). There was no significant difference in the presence or absence of preoperative chemotherapy or postoperative chemotherapy between the two groups (Table 1).

Patients with large tumors (tumor size of >3.1 cm) had significantly more lymph node metastases, lymphatic invasion, venous invasion, duodenal invasion, and infiltration into the anterior and posterior pancreatic tissue and portal vein system (S1 Table in S1 File). Finally, the survival rate was significantly worse in the ER group than in the non-ER group (p < 0.001) (Fig 1).

## Derivation of cut-off points of PLR, CA19-9 and tumor size for ER prediction

Focusing on preoperative factors, we used ROC curve analyses to evaluate the diagnostic performance of the PLR, CA19-9, and tumor size for ER. In the assessment of the PLR for ER prediction, the AUC, cut-off point, sensitivity, and specificity were 0.582, 0.467, 98.2%, and 20.2%, respectively (Fig 2A, Table 2). Similarly, in the assessment of the CA19-9 concentration for ER prediction, the AUC, cut-off point, sensitivity, and specificity were 0.686, 42 U/mL, 72.2%, and 64.7%, respectively (Fig 2B, Table 2). In the assessment of the tumor size for ER prediction, the AUC, cut-off point, sensitivity, and specificity were 0.642, 3.1 cm, 59.3%, and 71.7%, respectively (Fig 2C, Table 2).

## PLR, CA19-9 and tumor size as independent predictors of ER of resected PDAC

All continuous parameters in Table 3 were divided into two groups using ROC analysis, followed by univariate and multivariate analysis. The univariate analysis showed that a high PLR (p = 0.0125), high CA19-9 concentration (p < 0.0001), high SPan-1 concentration (p < 0.0001), tumor size of >3.1 cm (p = 0.0002), and lymph node metastasis (p = 0.0192)

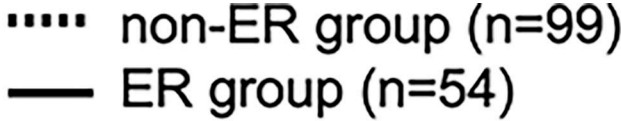

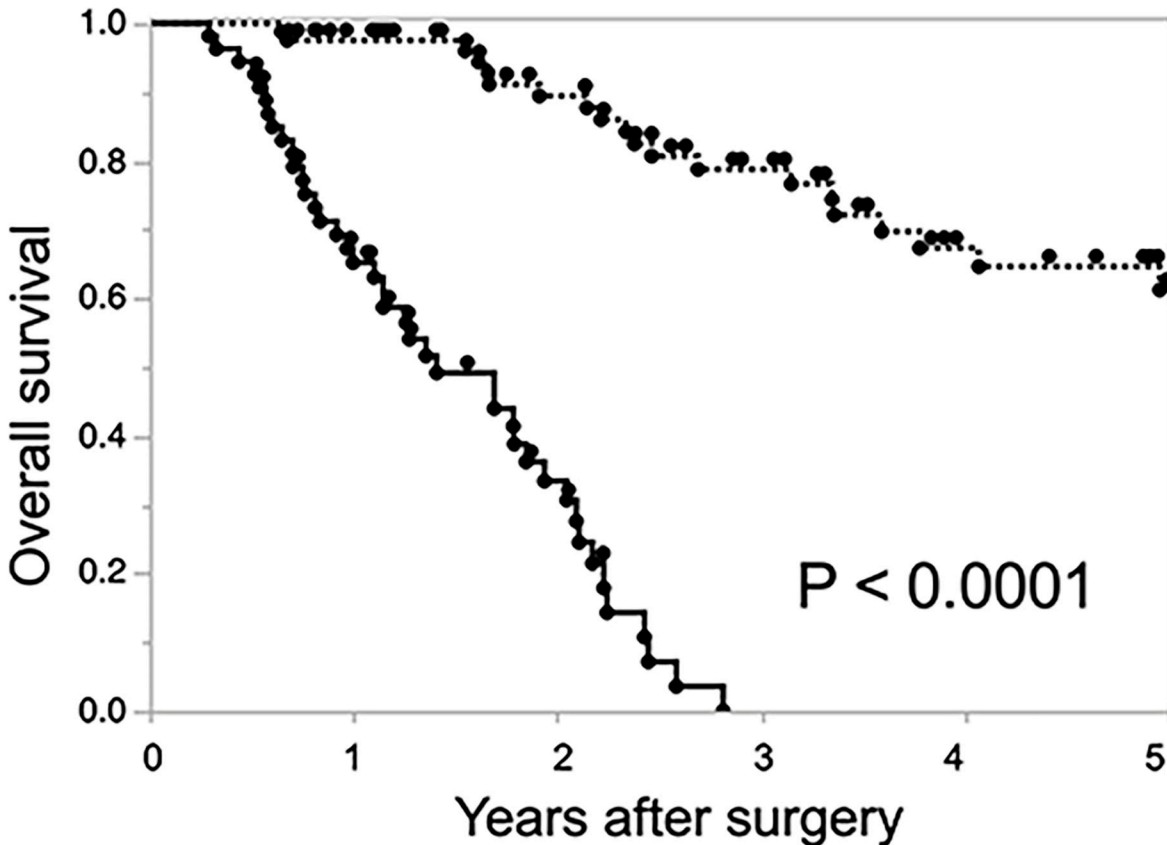

**Fig 1. Kaplan–Meier analysis of overall survival in patients who underwent pancreatectomy for pancreatic ductal adenocarcinoma divided into the ER group (n = 54) and non-ER group (n = 99).** ER, early recurrence.

were significantly associated with ER after resection of PDAC (Table 3). The multivariate analysis confirmed that a high PLR (odds ratio, 10.72; 95% confidence interval, 1.27–90.5; p = 0.0293), high CA19-9 concentration (odds ratio, 2.79; 95% confidence interval, 1.17–6.62; p = 0.0204), and tumor diameter of >3.1 cm (odds ratio, 2.42; 95% confidence interval, 1.11–5.26; p = 0.0261) were independently associated with ER in patients who underwent pancreatectomy for PDAC (Table 3).

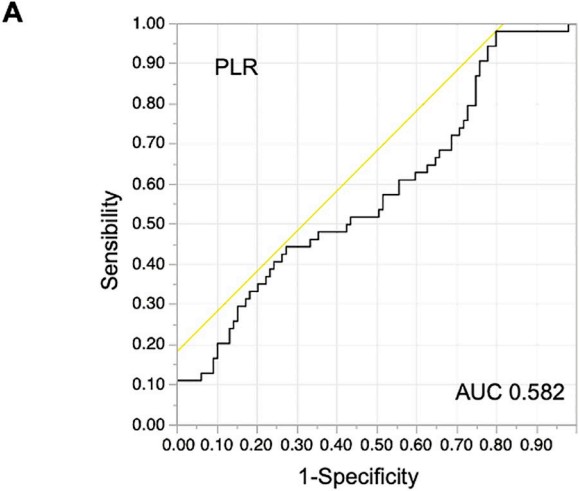

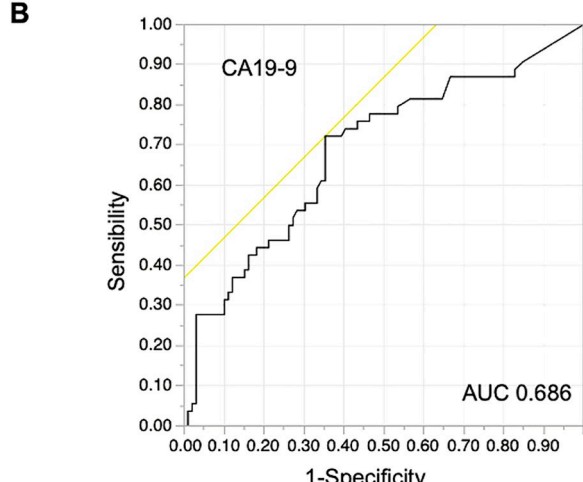

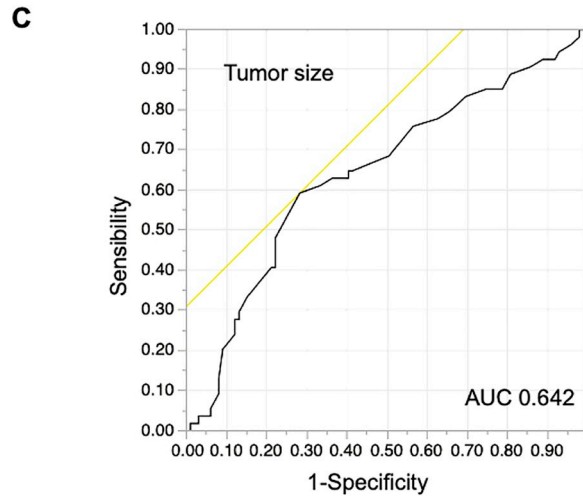

**Fig 2.** Receiver operating characteristic curves evaluating the accuracy of the (A) PLR, (B) CA19-9 concentration and (C) tumor size for the prediction of early recurrence after resection. PLR, platelet-to-lymphocyte ratio; CA19-9, carbohydrate antigen 19–9; AUC, area under the curve.

**Table 2. Determination of cut-off values for categorizing patients into ER or non-ER group (n = 153).**

|  | AUC | Cut-off point | Sensitivity (%) | Specificity (%) |
|---|---|---|---|---|
| PLR | 0.582 | 0.467 | 98.2 | 20.2 |
| CA19-9 (U/mL) | 0.686 | 42 | 72.2 | 64.7 |
| Tumor size (cm) | 0.642 | 3.1 | 59.3 | 71.7 |
| **The scoring system** | **0.757** | **2** | **87.0** | **52.5** |
| **The ER prediction model** | **0.763** | **3.3** | **85.2** | **55.6** |

ER, early recurrence; AUC, area under the receiver operating characteristic curve; PLR, platelet-to-lymphocyte ratio; CA19-9, carbohydrate antigen 19–9.

The diagnostic performance of each parameter for ER of resected PDAC was evaluated in 153 patients. The optimal cut-off point was determined as the number yielding the minimal value for $(1 - \text{sensitivity})^2 + (1 - \text{specificity})^2$ so that the sensitivity and specificity values were the closest to (0, 1) on the receiver operating characteristic curve.

## New scoring system using perioperative predictors

We stratified the ER prediction score from 0 to 3 points by calculating three preoperative prognostic factors (PLR, CA19-9, and tumor size) with assignment of 1 point for each factor. When each parameter exceeded the cut-off level, 1 point was given, and the total score of the three factors was defined as the ER prediction score. The total score of the identified factors was taken as the ER prediction score (0 points, n = 16; 1 point, n = 43; 2 points, n = 59; and 3 points, n = 35).

When DFS was compared by the ER prediction score, a higher score was associated with a worse prognosis of DFS (Fig 3A). Patients with a higher score (score of 2 or 3) had poorer DFS

**Table 3. Univariate and multivariate logistic regression analyses for ER.**

| Variable | Univariate analysis | | | Multivariate analysis | | |
|---|---|---|---|---|---|---|
|  | OR | 95% CI | *P value* | OR | 95% CI | *P value* |
| Gender (male) | 1.47 | 0.74–2.92 | 0.2671 |  |  |  |
| Age (≥63 vs < 63) | 1.98 | 0.93–4.18 | 0.0749 |  |  |  |
| BMI (≥20.04 vs < 20.04) | 1.74 | 0.72–4.21 | 0.2193 |  |  |  |
| DM (yes/no) | 1.07 | 0.55–2.08 | 0.8420 |  |  |  |
| Alb (≥4.5 vs < 4.5) | 0.26 | 0.05–1.46 | 0.1248 |  |  |  |
| P-Amy (≥81 vs < 81) | 1.65 | 0.66–4.11 | 0.2843 |  |  |  |
| NLR (≥1.231 vs < 1.231) | 2.41 | 0.76–7.61 | 0.1341 |  |  |  |
| **PLR (≥0.467 vs < 0.467)** | **13.42** | **1.75–103.0** | **0.0125** | **10.72** | **1.27–90.5** | **0.0293** |
| LMR (≥5.672 vs < 5.672) | 0.40 | 0.27–1.04 | 0.0778 |  |  |  |
| Preoperative chemotherapy (yes/no) | 0.64 | 0.32–1.28 | 0.2071 |  |  |  |
| CEA (≥33 vs < 33) | 1.09 | 0.10–12.33 | 0.9428 |  |  |  |
| **CA19-9 (≥42 vs < 42)** | **4.75** | **2.30–9.81** | **<0.0001** | **2.79** | **1.17–6.62** | **0.0204** |
| DUPAN-2 (≥66 vs < 66) | 1.52 | 0.85–3.46 | 0.1049 |  |  |  |
| SPan-1 (≥30 vs < 30) | 4.00 | 1.98–8.08 | **<0.0001** | 1.65 | 0.69–3.94 | 0.2558 |
| **Tumor size (≥3.1 vs < 3.1)** | **3.69** | **1.84–7.40** | **0.0002** | **2.42** | **1.11–5.26** | **0.0261** |
| Lymph node metastasis (yes/no) | 2.38 | 1.15–4.92 | **0.0192** | 1.44 | 0.63–3.27 | 0.3850 |
| Differentiation (poorly) | 1.67 | 0.84–3.28 | 0.1403 |  |  |  |
| Adjuvant chemotherapy (yes/no) | 0.93 | 0.40–2.13 | 0.8607 |  |  |  |

ER, early recurrence; OR, odds ratio; CI, confidence interval; BMI, body mass index; DM, diabetes mellitus; Alb, albumin; P-Amy, pancreatic amylase; NLR, neutrophil-to-lymphocyte ratio; PLR, platelet-to-lymphocyte ratio; LMR, lymphocyte-to-monocyte ratio; CEA, carcinoembryonic antigen; CA19-9, carbohydrate antigen 19–9. Boldface p-values are statistically significant.

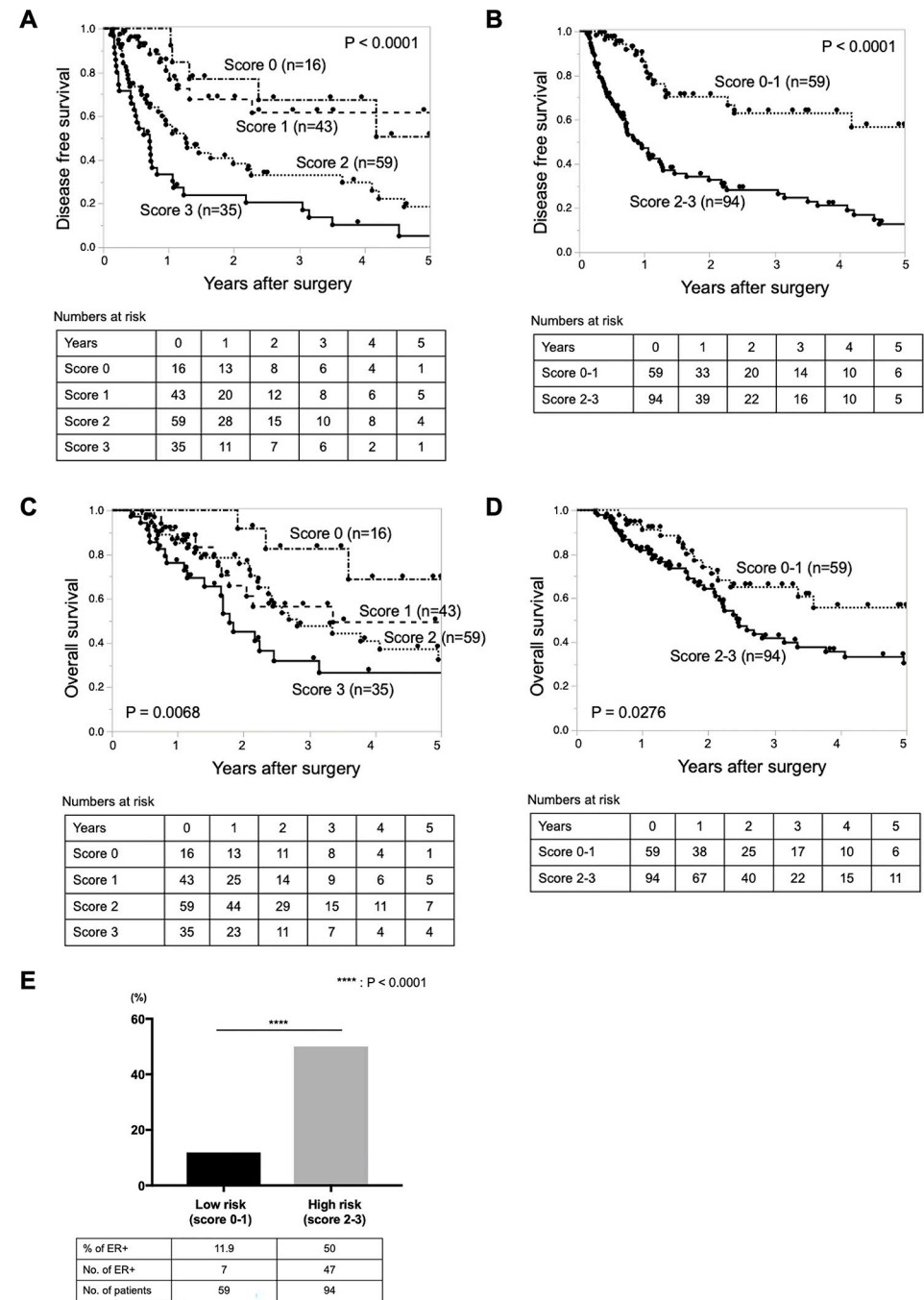

**Fig 3.** Kaplan–Meier analysis of (A, B) disease-free survival and (C, D) overall survival of patients who underwent pancreatectomy for pancreatic ductal adenocarcinoma stratified by the prediction scoring system. (E) ER developed in 7 (11.9%) patients in the low-risk group and in 47 (50.0%) patients in the high-risk group. ER, early recurrence.

than those with a lower score (score of 0 or 1) (Fig 3B). Similarly, when OS was compared by the ER prediction score, a higher score was associated with a worse prognosis of OS (Fig 3C). Patients with a higher score (score of 2 or 3) had poorer OS than those with a lower score (score of 0 or 1) (Fig 3D). The percentage of patients with ER was significantly higher among patients with higher scores (score of 2 or 3) than among those with lower scores (score of 0 or 1) (50.0% vs. 11.9%, respectively; p < 0.0001) (Fig 3E).

We performed this scoring system analysis in total 153 cases combining 55 NAT cases and 98 upfront surgery cases, using parameters measured before NAT (at time of first diagnosis). As a result, the significant worse prognosis for both OS and DFS were observed in the cases with higher score (S2A-S2D Fig in S1 File), which was similar to the main results (Fig 3A–3D).

We also did this scoring system analysis only in 55 cases that underwent NAT. We compared OS and DFS stratified by the prediction scoring system using parameters before and after NAT (S3A-S3D Fig in S1 File). As a result, both OS and DFS tended to be worse in the cases with higher score, whether the parameters before or after NAT were used. There was no statistically significant difference due to small number of cases. These results indicated that the preoperative parameters have the same significance even if it is measured before or after preoperative chemotherapy.

## The ER rates in every score of new scoring system

Using this new scoring system, the ER rate was calculated to be 0% (0 /16 patients) in the 0-point group, 16.2% (7 /36 patients) in the 1-point group, 40.7% (24 /59 patients) in the 2-point group, and 65.7% (23 /35 patients) in the 3-point group. The higher the score was, the higher the ER rate was. Then, SSLR was calculated to be 0 in the 0-point group, 0.356 in the 1-point group, 1.257 in the 2-point group, and 3.513 in the 3-point group.

## ROC analyses of new scoring system for ER of resected PDAC

When the ROC curve was examined for ER using this new scoring system, the scoring system was found to be accurate in diagnosing ER with an AUC of 0.757, cut-off point of 2, sensitivity of 87%, and specificity of 52.5% (Fig 4, Table 2).

## Developing an ER prediction model using high PCR and high CA19-9 and tumor diameter $>$ 3.1cm

Next, we calculated a predictive model for ER after the resection for PDAC by stepwise multiple logistic regression analysis as follows. If a patient had a high PLR, a high CA19-9, or a tumor diameter $>$ 3.1 cm, they received 1 point each in this model.

$$\text{Logit}(p) = 1.6 + 1.2 \times \text{high PLR} + 0.7 \times \text{high CA19-9} + 0.5 \times \text{tumor diameter} > 3.1\text{cm}$$

The AUC, sensitivity, and specificity of our prediction model were 0.763, 85.2% and 55.6%, respectively (Fig 4, Table 2). This ER prediction model was comparable to the scoring system and more useful than any test for identifying ER after the resection of PDAC (Table 2).

## Discussion

In the present study, we examined several perioperative parameters to predict ER and the prognosis after surgical resection of PDAC: (1) nutritional and immunological prognostic factors (NLR, PLR, lymphocyte-to-monocyte ratio, mGPS, total lymphocyte count, prognostic nutritional index, CONUT score, and C-reactive protein/albumin ratio), (2) tumor markers, and (3) pathological evaluation findings. Our results showed that a high PLR, high CA19-9 concentration, and large tumor size were reliable predictive markers for ER within 1 year after surgery and a poor prognosis, and it was possible to stratify postoperative DFS and OS depending on the ER prediction score.

The definition of ER after surgery for PDAC varies among previous reports, having been defined as 6 months [18], 8 months [19], and 1 year [20]. Groot et al. [21] classified patients with PDAC based on the postoperative recurrence time monthly from within 3 months to

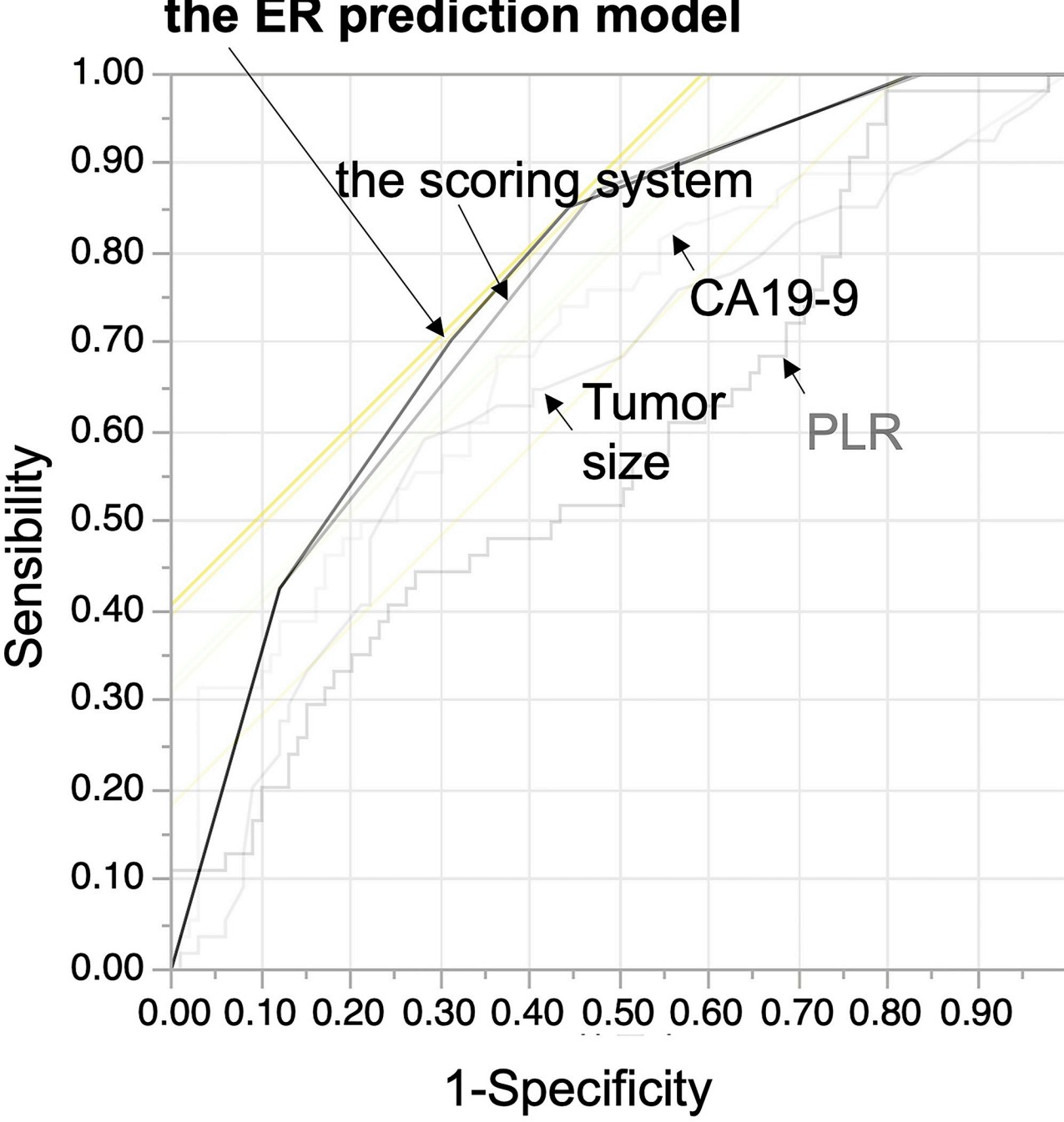

**Fig 4. Receiver operating characteristic curves evaluating the accuracy of the new scoring system and the ER prediction model for the prediction of early recurrence after resection.** ER, early recurrence; PLR, platelet-to-lymphocyte ratio; CA19-9, carbohydrate antigen 19–9; AUC, area under the curve.

within 20 months and evaluated the difference in prognosis between early and late recurrence focusing on the p-value of the survival curve analysis. They showed that the postoperative ER cut-off value that showed the greatest difference in prognosis was 1 year [21]. They also reported that adjuvant therapy was an independent factor that reduced the risk of ER [21]. Therefore, in the present study, ER was set at 1 year after surgery.

Malnutrition is a frequent and serious problem in patients with various cancers, and in many patients, malnutrition negatively affects patients' quality of life and prognosis [22]. Additionally, systemic inflammation, which is associated with disturbance of various hematological components such as white blood cells (specifically neutrophils, lymphocytes, and monocytes) and platelets, plays a critical role in cancer progression [23].

The NLR and the PLR have been reported in many nutritional evaluation methods using blood cell components, and they have a mutually complementary relationship. Neutrophils and platelets are representative blood cell components responsible for inflammatory reactions, and thrombocytosis is often observed in patients who have solid cancer with chronic inflammation [24, 25]. In addition, platelets themselves are deeply involved in cancer progression [26], and lymphocytes play an important role in cell-mediated immunity by initiating a cytotoxic immune response and by inhibiting cancer cell proliferation, invasion, and migration [23]. The PLR, which is the ratio of the platelet count to lymphocyte count, is a nutritional index that combines inflammatory and immunity indexes and serves as a reasonable marker for predicting cancer progression [27]. In fact, the preoperative PLR has been shown to be useful for prognostic prediction in various gastrointestinal cancers, and in recent years, the possibility of predicting the effect of systemic chemotherapy has also been suggested [27].

CA19-9 antigen is a sugar chain antigen derived from cell membrane glycolipids. It is expressed not only in colorectal cancer but also in other gastrointestinal cancers such as PDAC and cholangiocarcinoma [28, 29]. The CA19-9 concentration directly reflects the malignant potential of cancer and is reportedly a predictive marker for tumor staging, prognosis, and the response to chemotherapy in patients with PDAC [7, 28, 30]. Like the present study, previous studies have revealed an association of cut-off values of the preoperative serum CA19-9 concentration with ER and a poor prognosis [8, 30].

In addition, the tumor size was a parameter associated with ER after PDAC resection. As the tumor size increases, the likelihood of vascular invasion such as portal vein or superior mesenteric artery invasion increases, and vascular invasion is associated with a risk of subsequent multiorgan metastasis via blood vessels [31]. In the present study, the pathological results showed that the tumor diameter was significantly larger in patients with portal vein system invasion (PV1, n = 29) than in those without invasion (PV0, n = 124) (3.7 ± 0.2 vs. 2.6 ± 0.1 cm, p < 0.0001) (S1 Table in S1 File).

In this study, the multivariate analysis confirmed that a high PLR (p = 0.0293), high CA19-9 concentration (p = 0.0204), and tumor diameter of >3.1 cm (p = 0.0261) were independently associated with ER in patients who underwent pancreatectomy for PDAC (Table 3) and therefore, we created a new scoring system to predict postoperative ER with a poor prognosis using the preoperative PLR, serum CA19-9 concentration, and tumor size. Each factor had a different Odds ratio, which might give it different statistical strength to ER. Since the purpose of this study was to explore useful and simple biomarkers for clinical practice, we developed this scoring system. In fact, these 3 factors were known parameters those were reported to be important for the prognosis and recurrence of patients with PDAC [27]. By giving each parameter 1 point, we were able to prove that ER actually can be predicted by this new scoring system. Then, considering their odds ratio, the equation for the ER prediction model had the best at predicting ER after the resection of PDAC.

Our findings might have clinical implications. The prognosis of chemotherapy has recently improved, and the median survival time of patients with resectable pancreatic cancer who receive adjuvant chemotherapy is 46.5 to 54.4 months [32, 33]. Therefore, patients with a high ER score (especially patients of advanced age) should be carefully considered for surgery, and patients with PDAC who have a high score using this new prediction scoring system might be recommended to initially undergo and continue chemotherapy even if they have technically

resectable disease. In this way, the new ER score can be applied to treatment policy decisions in clinical practice.

Our study had several limitations. First, it was a single-center retrospective study with single cohort. To prove the results of new scoring system, it is better to have an independent validation cohort in a next step. Second, the cut-off values for the preoperative PLR, serum CA19-9 concentration, and tumor size require validation in a future prospective study. Third, the retrospective nature of the study prevented comparison of blood data and imaging findings because of differences in the number of chemotherapy courses and the contents of chemotherapy. Fourth, when OS was confirmed by stratifying by the presence or absence of preoperative treatment and postoperative adjuvant chemotherapy, the prognosis tended to be good in both groups that received chemotherapy (S1A, S1B Fig in S1 File). In the present study, a clear significant difference in OS between these groups was not observed because the indications and regimens for chemotherapy were not standardized in this retrospective study design. Fifth, the PLR and CA19-9 concentration are affected not only by cancer but also by infectious diseases such as cholangitis and the presence of preoperative chemotherapy. Finally, we have clearly demonstrated the SSLR in each score. Considering the SSLR in each score, this score will be particularly useful for patients with a score of 0 (SSLR = 0), which accounts for only 16 out of the total 153 patients. However, for the majority of patients (137 out of 153), calculating this score did not significantly alter the probability of ER when compared to the overall ER rate (35%).

In conclusion, this study revealed that postoperative ER for PDAC can be predicted by the new scoring system using the preoperative PLR, serum CA19-9 concentration, and tumor size. This ER prediction model may have great significance in identifying patients with a poor prognosis and avoiding unnecessary surgery.

## Supporting information

**S1 File.**
(DOCX)

## Acknowledgments

The authors also thank Angela Morben, DVM, ELS, from Edanz (https://jp.edanz.com/ac), for editing a draft of this manuscript.

## Author Contributions

**Conceptualization:** Tomonari Shimagaki, Keishi Sugimachi.

**Data curation:** Tomonari Shimagaki, Yohei Mano, Emi Onishi.

**Formal analysis:** Tomonari Shimagaki, Takahiro Tomino, Yuichiro Nakashima, Masahiko Sugiyama, Manabu Yamamoto, Masaru Morita, Mototsugu Shimokawa, Tomoharu Yoshizumi.

**Investigation:** Yohei Mano, Emi Onishi.

**Supervision:** Yasushi Toh.

**Writing – original draft:** Tomonari Shimagaki.

**Writing – review & editing:** Keishi Sugimachi.

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
