## [Decision Letter · Decision Letter 0]

27 Mar 2023

PONE-D-23-03771A new scoring system with simple preoperative parameters as predictors of early recurrence of pancreatic ductal adenocarcinomaPLOS ONE

Dear Dr. Shimagaki,

Thank you for submitting your manuscript to PLOS ONE. After careful consideration, we feel that it has merit but does not fully meet PLOS ONE’s publication criteria as it currently stands. Therefore, we invite you to submit a revised version of the manuscript that addresses the points raised during the review process.

 Please submit your revised manuscript by May 11 2023 11:59PM. If you will need more time than this to complete your revisions, please reply to this message or contact the journal office at plosone@plos.org. Please include the following items when submitting your revised manuscript:A rebuttal letter that responds to each point raised by the academic editor and reviewer(s). You should upload this letter as a separate file labeled 'Response to Reviewers'.A marked-up copy of your manuscript that highlights changes made to the original version. You should upload this as a separate file labeled 'Revised Manuscript with Track Changes'.An unmarked version of your revised paper without tracked changes. You should upload this as a separate file labeled 'Manuscript'.

We look forward to receiving your revised manuscript.

Kind regards,

Academic Editor

PLOS ONE

Journal Requirements:

https://link.springer.com/article/10.1245/s10434-021-10866-6?

https://pubmed.ncbi.nlm.nih.gov/36338597/

https://www.sciencedirect.com/science/article/abs/pii/S1424390321005263?via%3Dihub

In your revision ensure you cite all your sources (including your own works), and quote or rephrase any duplicated text outside the methods section. Further consideration is dependent on these concerns being addressed.

Additional Editor Comments:

Please revise.

Reviewers' comments:

Reviewer's Responses to Questions

**Comments to the Author**

1. Is the manuscript technically sound, and do the data support the conclusions?

Reviewer #1: Partly

Reviewer #2: Yes

2. Has the statistical analysis been performed appropriately and rigorously? 

Reviewer #1: No

Reviewer #2: Yes

3. Have the authors made all data underlying the findings in their manuscript fully available?

Reviewer #1: Yes

Reviewer #2: Yes

4. Is the manuscript presented in an intelligible fashion and written in standard English?

Reviewer #1: Yes

Reviewer #2: Yes

5. Review Comments to the Author

Reviewer #1: To the Editors:

Thank you for the opportunity to review a manuscript entitled “A new scoring system with simple preoperative parameters as predictors of early recurrence of pancreatic ductal adenocarcinoma”.

This paper is an article which introduced the predictive score of early recurrence using the preoperative serum CA19-9, PLR, and tumor diameter. This manuscript has been reviewed. I think that it is necessary to verify whether the methodology of this statistical analysis is correct or not by statisticians. There seems to be several problem in this statistical analysis. Although the report appears to give some clinically significant information, there are several points that need to be clarified.

Reviewer #2: The study tried to make a new scoring system to predict early recurrence of PDAC. The authors tried to make it to be optimal to predict ER. These approaches were well-done. However, they did not evaluate if this scoring system is effective in other cohorts. Without this evaluation, it is hard to score this developed scoring system. Without it, this study is merely a confirmation to the clinicopathologic evaluation of PDAC reported by many groups.

The cohort used consisted of NAT (+) and NAT (-) cases. The two groups may differ clinico-pathologically and biologically (e.g., the same T factor but the cancer cell density is sometimes very different between NAT (+) and NAT (-) PDAC). If so, scoring system optimal to predict early recurrence of PDAC may be changed dependent on composition of PDAC with and without NAT in the cohort used. What do the authors think about this?

6. PLOS authors have the option to publish the peer review history of their article (what does this mean?). If published, this will include your full peer review and any attached files.

Reviewer #1: **Yes: **Yusuke Yamamoto

Reviewer #2: No

---

## [Author Response · Author response to Decision Letter 0]

22 May 2023

Answers to the Reviewer #1:

We appreciate for the thoughtful comments and suggestions. Our responses to them are as follows. We changed the manuscript accordingly.

Q1.

The title of Table 3 was “univariate and multivariate Cox regression analysis for ER”. Is this analysis really Cox regression analysis? Please check by the statisticians. If possible, please add the statisticians as one of the co-authors.

Our response to Q1:

We appreciated the reviewer’s comment and Table 3 has been modified for univariate and multivariate analysis using logistic regression analyses. We asked a statistician to review this study and included him as a co-author. 

The above is described in the materials and methods section (page 10, lines 13) and in Table 3.

Q2.

In the Table 3, only four consecutive parameters were divided into two groups using the ROC analysis. I think that it will not be appropriate to divide only four consecutive parameters, and it will be preferable to divide all consecutive parameters into two groups using the ROC analysis. I recommend that these analysis will be supervised by the statisticians.

Our response to Q2:

As suggested by the reviewer, under the guidance of a statistician, all continuous parameters in Table 3 were divided into two groups using ROC analysis, followed by univariate and multivariate analysis.

The above is described in the results section (page 13, lines 5) and in Table 3.

Q3.

Authors should show the exact ER rates in every scores. I think that the ER rates of the patients with score 2 or 3, which are majority of this subjects (102 patients/153 patients), are probably similar ER rate of the entire ER rate (35%). So, this score will not actually be useful in majority of the pancreatic cancer patients.

Our response to Q3:

Using this new scoring system, the ER rate was calculated to be 0% (0 /16 patients) in the 0-point group, 16.2% (7 /36 patients) in the 1-point group, 40.7% (24 /59 patients) in the 2-point group, and 65.7% (23 /35 patients) in the 3-point group. The higher the score was, the higher the ER rate was.

The above is described in the results section (page 15, lines 1).

Q4.

The stratum-specific likelifood ratio (SSLR) indicates by how much a given diagnostic test result will increase or decrease the pretest probability of the target disorder. LR greater than 10 or less than 0.1 generates large and conclusive changes from pretest to post-test probability. Thus SSLR is estimated according to the following formula: SSLR = (x1/n1)/(x0/n0), where x1 is the number of patients in the stratum with ER; n1 is the total number of patients with ER; x0 is the number of patients in the stratum without ER; n0 is the total number of patients without ER (ref; Teasdale G, Jennett B (1974) Assessment of coma and impaired consciousness: a practical scale. Lancet 2:81–84). I recommend the authors to estimate SSLR for each score, and please prove the actual utility of this score. If the SSLR was greater than 10 or less than 0.1, this score will have the conclusive utility.

Our response to Q4:

The stratum specific likelihood ratio (SSLR) is the probability of a given test result when the disease is present, divided by the probability of the same test result when the disease is absent. We determined these ratios by means of the formula SSLR = (x1/n1)/(x0/n0), where x1 is the number of patients in the stratum with ER; n1 is the total number of patients with ER; x0 is the number of patients in the stratum without ER; n0 is the total number of patients without ER.

SSLR was calculated to be 0 in the 0-point group and 0.356 in the 1-point group, 1.257 in the 2-point group, and 3.513 in the 3-point group. 

The above is described in the materials and methos section (page 10, lines 14) and the results section (page 15, lines 4).

Q5.

 I recommend the authors to develop a score using each standardized variable based on the regression coefficient of the logistic regression model, because the odds ratio of each parameters of the score was greatly different. Considering their odds ratio, the equation for the scoring system had better be calculated on the assumption that a patient receives about 3-4 points for high PLR, if a patient receives 1 point for each of high CA19-9 and the tumor diameter of >3.1 cm."

Our response to Q5:

We appreciate the reviewer’s comment, and added logistic regression model in the revised manuscript.

We calculated a predictive model for ER after the resection for PDAC by stepwise multiple logistic regression analysis as follows. If a patient had a high PLR, a high CA19-9, or a tumor diameter > 3.1 cm, they received 1 point each in this model.

Logit(p) = 1.6 + 1.2 × high PLR + 0.7 × high CA19-9 + 0.5 × tumor diameter > 3.1cm 

The AUC, sensitivity, and specificity of our prediction model were 0.763, 85.2% and 55.6%, respectively (Figure 4, Table 2). This ER prediction model was comparable to the scoring system and was more useful than any test for identifying ER after the resection of PDAC (Table 2).

The above is described in the abstract and the results section (page 15, lines 10).

 

Answers to the Reviewer #2:

We appreciate for your thoughtful comments and suggestions. Our responses to the comments are as follows.

Q.

The cohort used consisted of NAT (+) and NAT (-) cases. The two groups may differ clinico-pathologically and biologically (e.g., the same T factor but the cancer cell density is sometimes very different between NAT (+) and NAT (-) PDAC). If so, scoring system optimal to predict early recurrence of PDAC may be changed dependent on composition of PDAC with and without NAT in the cohort used. What do the authors think about this?

Our response to Q:

We performed this scoring system analysis in total 153 cases combining 55 NAT cases and 98 upfront surgery cases, using parameters measured before NAT (at time of first diagnosis). As a result, the significant worse prognosis for both OS and DFS were observed in the cases with higher score (Supplementary Figure 2A-2D), which was similar to the main results (Figure 3A-3D). 

We also did this scoring system analysis only in 55 cases that underwent NAT. We compared OS and DFS stratified by the prediction scoring system using parameters before and after NAT (Supplementary Figure 3A-3D). As a result, both OS and DFS tended to be worse in the cases with higher score, whether the parameters before or after NAT were used. There was no statistically significant difference due to small number of cases. These results indicated that the preoperative parameters have the same significance even if it is measured before or after preoperative chemotherapy.

The above is described in the results section (page 14, lines 9).

---

## [Decision Letter · Decision Letter 1]

1 Jun 2023

PONE-D-23-03771R1A new scoring system with simple preoperative parameters as predictors of early recurrence of pancreatic ductal adenocarcinomaPLOS ONE

Dear Dr. Shimagaki,

Thank you for submitting your manuscript to PLOS ONE. After careful consideration, we feel that it has merit but does not fully meet PLOS ONE’s publication criteria as it currently stands. Therefore, we invite you to submit a revised version of the manuscript that addresses the points raised during the review process. Please submit your revised manuscript by Jul 16 2023 11:59PM. If you will need more time than this to complete your revisions, please reply to this message or contact the journal office at plosone@plos.org. Please include the following items when submitting your revised manuscript:A rebuttal letter that responds to each point raised by the academic editor and reviewer(s). You should upload this letter as a separate file labeled 'Response to Reviewers'.A marked-up copy of your manuscript that highlights changes made to the original version. You should upload this as a separate file labeled 'Revised Manuscript with Track Changes'.An unmarked version of your revised paper without tracked changes. You should upload this as a separate file labeled 'Manuscript'.If applicable, we recommend that you deposit your laboratory protocols in protocols.io to enhance the reproducibility of your results. Protocols.io assigns your protocol its own identifier (DOI) so that it can be cited independently in the future. For instructions see: https://journals.plos.org/plosone/s/submission-guidelines#loc-laboratory-protocols. Additionally, PLOS ONE offers an option for publishing peer-reviewed Lab Protocol articles, which describe protocols hosted on protocols.io. Read more information on sharing protocols at https://plos.org/protocols?utm_medium=editorial-email&utm_source=authorletters&utm_campaign=protocols.

We look forward to receiving your revised manuscript.

Kind regards,

Academic Editor

PLOS ONE

Journal Requirements:

**Additional Editor Comments:**

Please revise.

Reviewers' comments:

Reviewer's Responses to Questions

**Comments to the Author**

1. If the authors have adequately addressed your comments raised in a previous round of review and you feel that this manuscript is now acceptable for publication, you may indicate that here to bypass the “Comments to the Author” section, enter your conflict of interest statement in the “Confidential to Editor” section, and submit your "Accept" recommendation.

Reviewer #1: All comments have been addressed

Reviewer #2: All comments have been addressed

2. Is the manuscript technically sound, and do the data support the conclusions?

Reviewer #1: Yes

Reviewer #2: Yes

3. Has the statistical analysis been performed appropriately and rigorously? 

Reviewer #1: Yes

Reviewer #2: I Don't Know

4. Have the authors made all data underlying the findings in their manuscript fully available?

Reviewer #1: Yes

Reviewer #2: Yes

5. Is the manuscript presented in an intelligible fashion and written in standard English?

Reviewer #1: Yes

Reviewer #2: Yes

6. Review Comments to the Author

Reviewer #1: The authors have submitted a revised article on a new scoring system with simple preoperative parameters as predictors of early recurrence of pancreatic ductal adenocarcinoma. The data and methods have been thoroughly revised, and the results of this study have been clarified, offering important insights. I recommend making an additional correction as follows.

1. The authors have clearly demonstrated the SSLR in each score. Considering the SSLR in each score, this score will be particularly useful for patients with a score of 0 (SSLR = 0), which accounts for only 16 out of the total 153 patients. However, for the majority of patients (137 out of 153), calculating this score did not significantly alter the probability of early recurrence when compared to the overall early recurrence rate (35%). The authors should mention this limitation of the study in the limitations paragraph within the Discussion section.

Reviewer #2: The authors responded to the comment of the reviewer #2 and the revised manuscript has been improved.

7. PLOS authors have the option to publish the peer review history of their article (what does this mean?). If published, this will include your full peer review and any attached files.

Reviewer #1: No

Reviewer #2: No

---

## [Author Response · Author response to Decision Letter 1]

2 Jun 2023

Answers to the Reviewer #1:

We appreciate for the thoughtful comments and suggestions. Our responses to them are as follows. We changed the manuscript accordingly.

Q1.

The authors have clearly demonstrated the SSLR in each score. Considering the SSLR in each score, this score will be particularly useful for patients with a score of 0 (SSLR = 0), which accounts for only 16 out of the total 153 patients. However, for the majority of patients (137 out of 153), calculating this score did not significantly alter the probability of early recurrence when compared to the overall early recurrence rate (35%). The authors should mention this limitation of the study in the limitations paragraph within the Discussion section.

Our response to Q1:

We appreciated the reviewer’s comment. 

We have clearly demonstrated the SSLR in each score. Considering the SSLR in each score, this score will be particularly useful for patients with a score of 0 (SSLR = 0), which accounts for only 16 out of the total 153 patients. However, for the majority of patients (137 out of 153), calculating this score did not significantly alter the probability of ER when compared to the overall ER rate (35%).

The above is described in the discussion section (page 19, lines 14).

---

## [Decision Letter · Decision Letter 2]

19 Jun 2023

A new scoring system with simple preoperative parameters as predictors of early recurrence of pancreatic ductal adenocarcinoma

PONE-D-23-03771R2

Dear Dr. Shimagaki,

We’re pleased to inform you that your manuscript has been judged scientifically suitable for publication and will be formally accepted for publication once it meets all outstanding technical requirements.

Kind regards,

Academic Editor

PLOS ONE

Additional Editor Comments (optional):

Reviewers' comments:

Reviewer's Responses to Questions

**Comments to the Author**

1. If the authors have adequately addressed your comments raised in a previous round of review and you feel that this manuscript is now acceptable for publication, you may indicate that here to bypass the “Comments to the Author” section, enter your conflict of interest statement in the “Confidential to Editor” section, and submit your "Accept" recommendation.

Reviewer #1: All comments have been addressed

Reviewer #2: All comments have been addressed

2. Is the manuscript technically sound, and do the data support the conclusions?

Reviewer #1: Yes

Reviewer #2: Yes

3. Has the statistical analysis been performed appropriately and rigorously? 

Reviewer #1: Yes

Reviewer #2: Yes

4. Have the authors made all data underlying the findings in their manuscript fully available?

Reviewer #1: Yes

Reviewer #2: Yes

5. Is the manuscript presented in an intelligible fashion and written in standard English?

Reviewer #1: Yes

Reviewer #2: Yes

6. Review Comments to the Author

Reviewer #1: TO THE AUTHORS:

The authors have submitted a revised article on a new scoring system with simple preoperative parameters as predictors of early recurrence of pancreatic ductal adenocarcinoma. The data and methods have been thoroughly revised, and the study's results have been well clarified.

Reviewer #2: (No Response)

7. PLOS authors have the option to publish the peer review history of their article (what does this mean?). If published, this will include your full peer review and any attached files.

Reviewer #1: No

Reviewer #2: No

---

## [Editor Report · Acceptance letter]

7 Jul 2023

PONE-D-23-03771R2 

A new scoring system with simple preoperative parameters as predictors of early recurrence of pancreatic ductal adenocarcinoma 

Dear Dr. Shimagaki:

I'm pleased to inform you that your manuscript has been deemed suitable for publication in PLOS ONE. Congratulations! Your manuscript is now with our production department. 

Kind regards, 

on behalf of

Dr. Robert Jeenchen Chen 

Academic Editor

PLOS ONE